# Standing Balance Control of a Bipedal Robot Based on Behavior Cloning

**DOI:** 10.3390/biomimetics7040232

**Published:** 2022-12-09

**Authors:** Jae Hwan Bong, Suhun Jung, Junhwi Kim, Shinsuk Park

**Affiliations:** 1Department of Human Intelligence Robot Engineering, Sangmyung University, Cheonan-si 31066, Republic of Korea; 2Center for Healthcare Robotics, Korea Institute of Science and Technology, Seoul 02792, Republic of Korea; 3Innovation Center, Samsung Electronics Co., Ltd., Hwaseong-si 16678, Republic of Korea; 4Department of Mechanical Engineering, Korea University, Seoul 02841, Republic of Korea

**Keywords:** biped robots, robot motion, intelligent robots, robot learning, behavior cloning

## Abstract

Bipedal robots have gained increasing attention for their human-like mobility which allows them to work in various human-scale environments. However, their inherent instability makes it difficult to control their balance while they are physically interacting with the environment. This study proposes a novel balance controller for bipedal robots based on a behavior cloning model as one of the machine learning techniques. The behavior cloning model employs two deep neural networks (DNNs) trained on human-operated balancing data, so that the trained model can predict the desired wrench required to maintain the balance of the bipedal robot. Based on the prediction of the desired wrench, the joint torques for both legs are calculated using robot dynamics. The performance of the developed balance controller was validated with a bipedal lower-body robotic system through simulation and experimental tests by providing random perturbations in the frontal plane. The developed balance controller demonstrated superior performance with respect to resistance to balance loss compared to the conventional balance control method, while generating a smoother balancing movement for the robot.

## 1. Introduction

As robot technologies advance, various types of mobile robots, such as wheeled, legged, and tracked robots, are being investigated for a variety of applications. Many robotic applications are designed for human-scale environments, in which both robots and humans can physically coexist. A bipedal humanoid robot is considered the prime candidate for robotic tasks in human-scale spaces because its shape and dimensions are the same as those of the humans for which the space was designed. However, although remarkable advances have been made in humanoid technologies, humanoid robots with human-level mobility and manipulability are still a distant prospect. As demonstrated at the DARPA Robotics Challenge held in 2015, state-of-the-art robotic systems struggle with simple physical tasks that their human counterparts can easily perform in daily life [1].

Whereas a conventional industrial robot performs tasks on a fixed base, a bipedal humanoid robot is an underactuated multibody system on a floating base. Owing to the inherent instability of floating-base robots, it is difficult to control the balance of these humanoids while they perform locomotion and object manipulation.

Humanoid balance control has been investigated in two categories: walking balance [2,3] and standing balance. For their stable object manipulation and physical interaction with environment, the standing balance of humanoids has been investigated based on position control [4,5] and force control [6,7,8,9,10,11,12].

For standing balance control, force control methods are known to have several advantages over position control, by directly controlling the contact forces and moments by utilizing the low impedance of the humanoid mechanism even with a relatively small number of sensors. Several studies have been conducted based on passivity theory [6,7,8,9,10] for computing the joint torques necessary to generate the contact forces, which in turn yields the desired wrench (force and moment) at the center of mass (CoM) of the humanoid. Using the passivity theory approach, the humanoid can maintain balance while physically contacting with the environment at multiple contact points by producing the proper wrench at the CoM. For example, by setting the desired wrench at the CoM, the contact force produced by the manipulation task of the upper body can be compensated for by generating the appropriate contact forces at the feet. Simple feedback laws [6,7,8] and model predictive control (MPC) [9,10] have been proposed to compute the desired wrench at the CoM.

These methods using the simple feedback law and the MPC set the desired wrench to be proportional to the difference between the current states of the robot (e.g., position, velocity, and momentum of the CoM) and a fixed reference configuration (FRC). However, this approach is rather simplistic compared to the balancing control employed by humans. In human motor control based on a human’s own experiences, complex neural activities allow the human to maintain balance by changing the reference configuration pertaining to the human’s surrounding environment, rather than by using an FRC.

Humans, unlike their humanoid counterparts, have the innate ability to maintain balance by shifting their CoM. Human balance control is carried out by combining various balance strategies [13]. Anticipatory postural adjustment, as one of human balance strategies, suggests that human balance is recovered in a feedforward manner by predicting and reacting to balance perturbations. This is analogous to the MPC approach used for humanoid balance control. In addition, humans recover their balance using reflex responses. This strategy utilizes the spring-like property of the neuromuscular system, as in the case of the simple feedback control approach used for humanoid balance control. Another strategy used for human balance control is pre-programmed reactions to postural perturbations. This balancing strategy is based on tacit knowledge and is regarded as originating from motor skills that are context-dependent and highly non-linear [14]. It involves personal cognitive factors, the goal of balancing, and one’s own experiences [15,16]. Although a number of studies have been conducted on pre-programmed motor skills for human balancing [17,18,19], these skills have not yet been implemented in humanoid control algorithms.

Recently, artificial intelligence (AI) technologies have demonstrated capabilities that approach or exceed the human level, in various fields of application. Machine learning is a subfield of AI, in which AI algorithms learn from training data to make predictions or decisions. Among machine learning techniques, deep neural networks (DNNs) have shown superior performance in handling nonlinear problems with high robustness against noise in the fields of computer vision [20], speech recognition [21], and video games [22]. In addition, DNNs have been successfully used to achieve sensor fusion and to control dynamic systems, such as robots and vehicles [23,24,25,26].

Reinforcement learning and imitation learning are commonly used to train DNNs for robotic applications. Reinforcement learning algorithms with well-designed reward functions have been shown to be effective for locomotion applications [27,28,29]. However, reinforcement learning requires a large number of trial-and-error experiments or computer simulations to learn the best action strategy. Imitation learning is commonly used when it is easier to make a human demonstrate the desired robotic behavior, for example, for robot manipulation tasks [30,31,32], than when designing an appropriate reward function, which is a more demanding task. Compared to reinforcement learning or model-based automatic control, imitation learning can be more effective because it allows for the learning of complex balancing strategies directly from human demonstrations.

This study developed a novel balance controller for a bipedal robot based on machine learning techniques and robot dynamics. Behavior cloning, one type of imitation learning, is used to mimic human balancing strategies to recover from unknown force perturbations. The unknown force perturbations considered as force applied to the bipedal robot by environment while object manipulation. To cope with the perturbations, the desired wrench at the CoM is computed using a behavior cloning model trained by human demonstrations of balancing. To achieve the desired wrench at the CoM, the joint torques are calculated based on the robot dynamics. In this study, each perturbation was modeled as the internal force applied by the upper body on the lower body, while the upper body performed manipulation tasks that required physical contact with the environment. The performance of the balance controller was validated using a bipedal lower-body robotic system by simulation and experimental tests under random perturbations in the frontal plane. The results show that the developed balance controller is capable of generating a smooth balancing motion against various types of perturbations, compared to the conventional balance controller.

The remainder of this paper is organized as follows. Section 2 discusses the humanoid balance control system and explains the proposed controller. Section 3 presents and analyzes the simulation and the experimental results. Section 4 presents concluding remarks.

## 2. Materials and Methods

### 2.1. Overview of Developed Balance Control System

The humanoid under balance control can be modeled as two parallel-coupled subsystems: the upper and lower bodies. The upper body is modeled as an impedance system for manipulation, and the lower body is modeled as a floating base system for balancing [33,34]. The manipulation task performed by the upper body causes perturbation of both the upper and lower bodies [35]. As shown in Figure 1a, a humanoid manipulation task generates an interaction force at the contact point between the upper body and the environment. This contact force in turn produces the internal force and moment between the upper and lower bodies as an action-reaction pair, which are equal in magnitude and opposite in direction (Figure 1b). In this study, we employed a bipedal lower-body robot (Figure 1c), Little HERMES (Biomimetics Robotics Lab., MIT, Cambridge, MA, USA) [36], to test the developed balance controller through a series of simulations and experiments. In the balancing simulations and experiments, external forces were applied at random points on the trunk of the lower-body robot (Figure 1c).

The external force can be considered to have the same effect as the equivalent internal force and moment at the interface between the upper and lower bodies (Figure 1c), as if it was caused by the perturbation by the contact manipulation task (see Equation (1)).
(1)$$F_{inter}=F_{ext}\;M_{inter}=r\times F_{ext}$$

Figure 2 illustrates a schematic of the humanoid balance control system developed in this study. The external force along with the ground reaction forces on the feet creates the resultant wrench at the CoM of the lower-body robot, as shown in the figure. The balance controller commands the joint torques of the robot based on the measured CoM states. The balance controller is composed of two components: a wrench estimator and a joint torque controller. The wrench estimator uses a behavior cloning model that was trained using data from human-operated balancing. The human demonstration data were acquired from a human operator who used a force-reflecting human–machine interface (HMI) to teleoperate the balance of the bipedal robot. After training, the behavior cloning model learned a mapping from the measured robot states (Figure 2a) to the desired wrench at the CoM of the robot (Figure 2b). Using the behavior cloning model, the wrench estimator predicts the desired wrench at the CoM to maintain balance based on the human demonstration data for balancing. Based on the prediction from the wrench estimator, the joint torque controller calculates the joint torques in both legs of the bipedal robot via robot dynamics (Figure 2c). The joint torques, in turn, produce the ground reaction forces (GRFs) required to achieve the desired wrench at the CoM to maintain balance.

### 2.2. Teleoperation System for Collecting Human Demonstration Data

To collect human demonstration data, we used a teleoperation system in which a human operator controlled the balance of a bipedal robot with the approval of the Deliberation Committee (KUIRB-2020-0277-01). While collecting the data, an actual robot and a robot simulator were interacted with the human operator in a human-in-the-loop manner to maintain the balance by using a custom-made HMI. The custom-made HMI, Balance Feedback Interface (BFI), was developed by the MIT Biomimetics Robotics Laboratory [34]. The BFI synchronizes the dynamics between the human operator and the robot to transfer the balancing strategy of the human operator to the robot.

#### 2.2.1. Bipedal Robot

The actual robot, Little HERMES (MIT Biomimetics Robotics Laboratory, Cambridge, MA, USA), and the simulator for Little HERMES were used to acquire the data and evaluate the performance of the developed balance control system. The design of Little HERMES has several features that simplify its robot dynamics. The bipedal robot has spherical feet, which makes each foot of the robot contact with the ground at a single point. At a single point of contact with the ground, the GRF has only three directional force components without any moment components. Each leg has three actuated degrees of freedom for hip flexion/extension, hip abduction/adduction, and knee flexion/extension. Six actuators for the two legs were placed in the body trunk using timing belts for the actuation of the knee joints, as shown in Figure 3a. This feature effectively reduces the inertia of the leg, and the total mass and inertia of the bipedal robot can be modelled to be concentrated at the CoM of the body trunk. Owing to these features of the bipedal robot, the desired wrench at the CoM divides into the GRF for each foot. The joint torques are calculated from the GRF using the Jacobian matrices for the two legs.

The sensors implemented in the actual robot include an inertial measurement unit (IMU) and six joint encoders. The IMU attached to the body trunk was used to measure the orientation and three-directional angular velocities of the body trunk. The joint encoders measure the joint angles of the two legs. The CoM states of the robot can be estimated based on measurements from the IMU and the joint encoder.

We developed a dynamic simulator for Little HERMES. In the simulator, a multi-body dynamics model of Little HERMES was programmed using the physics engine MuJoCo [37]. The model was programmed to have the same kinematic and dynamic structure as its physical version, as shown in Figure 3b. The balance controller was first applied to the multi-body dynamics model before being tested on an actual robot to prevent unexpected behavior and breakdown of the actual robot.

#### 2.2.2. Teleoperation System with Balance Feedback Interface

Figure 4 illustrates the human-operated balancing system using balance feedback interface (BFI). The BFI is composed of two actuated arms and a force plate. The two arms were designed to apply a force and moment on the body trunk of the human operator and track its translational and rotational motion. The force plate measures the position of the center of pressure (CoP) and the resultant GRF relative to the CoP.

Upon the application of an external force on the bipedal robot, the change in the balance state of the robot is reflected by the force and moment applied by the two arms of the BFI on the human operator. The feedback force and moment are applied on the trunk of the human operator in the frontal plane (red arrows in Figure 4). This feedback allows the human operator to perceive the difference between the balance states of the human and the robot, and the human operator reacts to the feedback force and moment by shifting their CoM and applying force on the force plate. These human reactions to maintain one’s balance are then captured by the net wrench at the CoM and the divergent components of motion (DCM) of the human operator. The captured human reactions are scaled and used to control the balancing motion of the robot.

In the frontal plane, the net wrench at the CoM of the human operator (***W**_H_*) contains two force components (x- and z-axes) and one moment component (y-axis). The two force components at the CoM are equivalent to those of the resultant GRF measured by the force plate. The moment at the CoM can be estimated based on the two force components of the resultant GRF and the relative position between the CoM and the CoP, as described in Equation (2). $M_{y,H}$ denotes the moment at the CoM of the human operator along the y-axis in Equation (2). The components of the resultant GRF along the x- and z-axes are denoted by $F_{z,H},\;F_{x,H}$. The position of the CoP along x-axis is given by $p_{x,H}$. $x_{H},\;z_{H}$ are the position of the CoM along x- and z-axes.
(2)$$M_{y,H}=-F_{z,H}\cdot \left(p_{x,H}-x_{H}\right)-F_{x,H}\cdot z_{H}$$

The DCM [38], which is also referred to as the capture point, reflects the unstable dynamics of the CoM and can be estimated from the measured position and velocity of the CoM, as can be seen in Equation (3). The DCM in the frontal plane has two linear components (x- and z-axes) and one angular component (y-axis).
(3)$$\xi_{i}=S_{i}+{\dot{S}}_{i}\cdot \sqrt{\frac{h}{g}}$$

Here,



x,z,y ∈i



$\xi_{i}$: DCM along the i-axis

$S_{i},\;{\dot{S}}_{i}$: Position and velocity of the CoM along the *i*-axis

h: Nominal height

g: Gravitational acceleration

As illustrated in Figure 4a,b, both the net wrench at the CoM of the human operator (*W**_H_*) and the DCM of the human operator (*ξ**_H_*) are scaled to provide the reference wrench (*W**_ref_*) and the reference DCM (*ξ**_ref_*) for the bipedal robot. The reference DCM (*ξ**_ref_*) is then compared with the measured DCM of the bipedal robot (*ξ**_R_*). The difference between the reference DCM (*ξ**_ref_*) and the scaled DCM of the bipedal robot (Figure 4c) is multiplied by the control gain ***K*** (Figure 4d). The control gain ***K*** was determined by trial and error by minimizing the error between the scaled DCM of the human operator and the scaled DCM of the bipedal robot. Then, the result is summed with the reference wrench (*W**_ref_*) to yield the desired wrench at the CoM of the bipedal robot (*W**_D_*). The computation of the desired wrench (*W**_D_*) to maintain the balance of the bipedal robot is expressed as in Equation (4).
(4)$$W_{D}=\left[\begin{array}{c} \alpha_{x}\\ \alpha_{z}\\ \alpha_{y} \end{array}\right]⊙W_{H}+\left[\begin{array}{c} K_{x}\\ K_{z}\\ K_{y} \end{array}\right]⊙\left(\left[\begin{array}{c} \beta_{x}\\ \beta_{z}\\ \beta_{y} \end{array}\right]⊙\xi_{H}-\left[\begin{array}{c} \gamma_{x}\\ \gamma_{z}\\ \gamma_{y} \end{array}\right]⊙\xi_{R}\right)$$

Here,

⊙: Element-wise multiplication (also known as the Hadamard product)

$W_{D}={\left[W_{x,D}\;W_{z,D}\;W_{y,D}\right]}^{T}$: Desired wrench at the CoM of the robot

$W_{H}={\left[W_{x,H}\;W_{z,H}\;W_{y,H}\right]}^{T}$: Net wrench at the CoM of the human operator

${\left[\alpha_{x}\;\alpha_{z}\;\alpha_{y}\right]}^{T}={\left[\frac{m_{R}\cdot d_{R}\cdot h_{H}}{m_{H}\cdot d_{H}\cdot h_{R}}\;\frac{m_{R}}{m_{H}}\;\frac{m_{R}\cdot d_{R}}{m_{H}\cdot d_{H}}\right]}^{T}$: Scale factors for the $W_{H}$

${\left[K_{x}K_{z}\;K_{y}\right]}^{T}={\left[40\;80\;30\right]}^{T}$: Control gains

$\xi_{H}={\left[\xi_{x,H}\;\xi_{z,H}\;\xi_{y,H}\right]}^{T}$: DCM of the human operator along each axis

${\left[\beta_{x}\;\beta_{z}\;\beta_{y}\right]}^{T}={\left[\frac{1}{d_{H}}\;\frac{1}{h_{H}}\;\frac{I_{H}}{m_{H}\cdot d_{H}\cdot h_{H}}\right]}^{T}$: Scale factors for the $\xi_{H}$

$\xi_{R}={\left[\xi_{x,R}\;\xi_{z,R}\;\xi_{y,R}\right]}^{T}$: DCM of the robot along each axis

${\left[\gamma_{x}\;\gamma_{z}\;\gamma_{y}\right]}^{T}={\left[\frac{1}{d_{R}}\;\frac{1}{h_{R}}\;\frac{I_{R}}{m_{R}\cdot d_{R}\cdot h_{R}}\right]}^{T}$: Scale factors for the $\xi_{R}$

$m_{i},\;d_{i},h_{i},I_{i}$: mass, distance between the two feet along the x-axis, nominal height, and moment of inertia around the CoM of the human operator (*i = H*) and robot (*i = R*)

With the desired wrench at the CoM ($W_{D}$), the GRFs on the feet of the bipedal robot are determined based on the robot dynamics. The joint torques in the legs are computed from the GRFs using Jacobian matrices, as shown in the following equation:(5)$$\left[\begin{array}{c} \tau_{l}\\ \tau_{r} \end{array}\right]=\left[\begin{array}{cc} J_{l}{}^{T} & 0\\ 0 & J_{r}{}^{T} \end{array}\right]\cdot \left[\begin{array}{c} F_{l}\\ F_{r} \end{array}\right]$$

Here,

$\tau_{l},\tau_{r}$: Amounts of torque for three motors in the left and right leg

$J_{l},\;J_{r}$: Jacobian matrices for the left and right legs

$F_{l},\;F_{r}$: Contact forces on the left and right feet

### 2.3. Balance Controller Trained by Human Demonstration

#### 2.3.1. Acquisition of Training Data from Human-Operated Balancing

Data were collected from the robot and the human operator during successful human-operated balance control tasks. The collected data were used to train the balance controller based on a behavior cloning model, which was designed to mimic the balancing skills of the human operator (see Section 2.3.2 for more details). The data from the bipedal robot were collected using the dynamic simulator for Little HERMES, as well as the actual robot system. In both the actual robot system and its dynamic simulator, one human subject (33-year-old male with the height of 185 m and the weight of 88 kg) operated the BFI to maintain the balance of the bipedal robot when external forces were applied to the robot in the frontal plane. When using the actual robot system, a human experimenter manually applied external forces. The applied forces had irregular shapes with different amplitudes, patterns, and durations for each trial. In the dynamic simulator, external forces with square-wave forms were applied; the square waves had various amplitudes, whereas the pulse width was maintained at 180 ms. The external forces were applied after the robot had fully recovered its balance from the previously applied force.

While collecting the human-operated balancing data, the loss of balance of the bipedal robot was discerned under the following conditions: (1) at least one of the robot’s feet lost contact with the ground, and the robot was no longer in the double support phase; (2) the DCM moved outside the support line formed between the two feet of the robot in contact with the ground; and (3) the angle between the midline of the robot trunk and the vertical line in the frontal plane exceeded 20°. If any of the three conditions were met, the bipedal robot was considered to have lost its balance, and data acquisition was halted.

Figure 5 describes the data collected from the robot and the human operator. Figure 5a shows the CoM states of the robot that serve as the inputs to the DNN model for the wrench estimator: the linear position and velocity of the CoM along the x- and z-axes, the angular position and velocity of the robot trunk around the y-axis, and the DCM along the x-, z-, and y-axes. Figure 5b shows the data acquired from the BFI, which was then scaled to serve as the target output of the DNN in the balance controller: the reference DCM and wrench ($\xi_{ref}$ and $W_{ref}$). As described in Equation (3), the human DCM and wrench ($\xi_{H}$ and $W_{H}$) were scaled by scaling factors ${\left[\beta_{x},\beta_{z},\;\beta_{y}\right]}^{T}$ and ${\left[\alpha_{x},\;\alpha_{z},\;\alpha_{y}\right]}^{T}$, respectively. The robot states and human-operated data were sampled every 1.8 ms.

#### 2.3.2. Balance Controller Trained by Human Demonstration Data

The balance controller was trained to control the balancing motion of the robot using the human-operated balancing data described in the previous section. Figure 6 illustrates a detailed schematic of the developed balance controller. The wrench estimator replaces the role of the human operator in controlling the balance of robots by estimating the desired wrench for the robot ($W_{D}$), as shown in the figure. As the input to the wrench estimator, the nine features of the robot states were accumulated over the past 100 time steps, including the most recent time step. The nine features of the robot states were identical to the data acquired from the robot to train the balance controller (see Figure 5a). The wrench estimator was developed based on a behavior cloning method. The behavior cloning method is a branch of imitation learning in which the algorithm learns a human-like policy from the data of human demonstrations. The behavior cloning model used in the wrench estimator has two separate networks: *DCM Net* and *W Net*.

The structure and hyperparameters of the DNN should be carefully determined before training the DNN using human demonstration data. Among the various DNN structures, long short-term memory (LSTM) is known to be suitable for handling time series data because it can model complex nonlinear feature interactions [39,40]. However, other studies have reported that multi-layer perceptrons (MLP) and convolutional neural networks (CNN) perform better than LSTM in processing time series data [41,42]. In this study, three types of DNN structures were tested as candidate structures for both DCM Net and W Net: MLP, CNN-MLP, and LSTM-MLP, where CNN-MLP and LSTM-MLP are hybrid algorithms of a CNN and LSTM with a MLP, respectively. The hyperparameters of the DNN are the variables to be tuned to optimize the DNN structure and learning process. Table 1 lists the types and ranges of the hyperparameters that were explored for the three types of DNN structures compared in this study. The Bayesian optimization method was used to explore the hyperparameters for each DNN structure. The Bayesian optimization method iteratively explores the predefined range of the hyperparameters (see Table 1). Since the Bayesian optimization method utilizes the results from the previous exploration to find better hyperparameters, it is known to converge faster compared to other hyperparameter tuning methods, such as grid search and random search, wherein an explored combination of hyperparameters is independent of other previously explored hyperparameters.

The best DNN structures of the *DCM Net* and *W Net* were selected by following the process described in Figure 7. The selection process was performed using data collected from both the actual robot-BFI system and the robot simulator-BFI system. Through iterative explorations of the hyperparameter space using the Bayesian optimization method, the hyperparameters of the three DNN structures (MLP, CNN-MLP, and LSTM-MLP) were optimized to minimize the root mean square error (RMSE) and the computation time. The RMSE indicates how closely the estimation by the DNN structure fits the human-operated balancing data. The computation time is the time required for the DNN structure to process the input data to yield the output, which is critical for the real-time control of the robotic system. The performances of the three DNN structures after optimization were compared with respect to their RMSE and computation time to determine the DNN structures for *DCM Net* and *W Net* (see Section 3.1 for the results).

After the optimization process of the hyperparameters, the *DCM Net* and the *W Net* were trained in a supervised manner using different pairs of input-target data. For training both networks, the nine features of the robot states and the reference DCM and wrench ($\xi_{ref}$ and $W_{ref}$) were used as input data and target data, respectively. The trained *DCM Net* and *W Net* mapped the robot states onto the reference DCM and wrench in a similar way to the regression algorithms of supervised learning. The behavior cloning method, however, is known to suffer from a compounding error that may cause the robot states to drift away from human demonstrations [43]. In this study, the behavior cloning model was combined with a feedback control method to compensate for the compounding error. As shown in Figure 6b, the current DCM of the robot ($\xi_{R}$) was fed back to the wrench estimator to compute the desired wrench ($W_{D}$), as described in Equation (3).

The desired wrench for the bipedal robot ($W_{D}$) was then fed to the joint torque controller. Under the condition of double support, the GRFs at the two feet were computed to produce the desired wrench that counteracted the external force [36], as shown in Figure 6d. The GRFs were mapped onto the joint torques for the two legs using contact Jacobian matrices (Figure 6e). The bipedal robot can maintain balance by applying a joint torque to compensate for the external force (Figure 6f). The sampling time for the control loop in Figure 6 was 1.8 ms, both for the actual robot and its dynamic simulator.

### 2.4. Simulation and Experimental Tests of Balance Controller

Most studies on humanoid balance have been based on an FRC [6,7,8,9,10]. In the studies using an FRC, the desired wrench at the CoM was computed to be proportional to the difference between the current configuration and the FRC of the robot, whereas the wrench estimator developed in this study computes the desired wrench at the CoM of the robot from the robot states by capturing human balancing skills without reference to an FRC.

The performance of the balance controller developed in this study was compared with that of a conventional balance controller using an FRC, as illustrated in Figure 8. In the conventional balance controller, the FRC is set as the upright posture of the robot, and the desired wrench is set to be proportional to the difference between the current configuration and the FRC. The joint torque is calculated in the same manner as that used in the developed balance controller.

#### 2.4.1. Simulation Tests with Robot Dynamics Simulator

To avoid robot balancing motions that may cause unpredicted damage to the actual bipedal robot, the developed balance controller was first tested using a dynamic simulator. In simulation tests, the performance of the developed balance controller was evaluated based on the resistance to balance loss and the jerk of the balancing motion. To evaluate the balancing performance in the frontal plane, the bipedal robot in the dynamic simulator was confined to move in the frontal plane, and external forces of square-wave forms were applied laterally to the CoM of the robot.

To evaluate the capability of the robot to resist balance loss, external forces were applied as a single pulse and as a series of pulses, which had different patterns from the external forces used for training the balance controller. For the simulations, five values of the pulse widths were provided: 180, 150, 100, 50, and 10 ms. The pulse amplitude was increased from 0 N with an increment of 1 N until one foot of the robot lost contact with the ground. The capability of the robot to resist balance loss was assessed by monitoring the amplitude of the external force at which either foot slip or loss of foot-ground contact occurred. A higher force amplitude indicated a greater resistance to balance loss. In the case of the simulation using the series of pulses, the pulses were applied at intervals of 400 ms, so that the pulse was applied before the robot fully recovered its balance from the previous pulse.

The jerkiness of the balancing motion was assessed by estimating the jerk cost function of the CoM of the robot while applying a single pulse of external force with a pulse width of 100 ms. The acceleration at the CoM was passed through a low pass filter of 10 Hz, eliminating the higher frequency noise. The cut-off frequency was chosen to be 10 Hz because the typical frequency of the body motion is often observed below 10 Hz [44]. The jerk at the CoM of the robot was computed by numerically differentiating the filtered acceleration at the CoM. The jerk cost functions of the linear and angular movements were calculated as follows:(6)$$J_{L}=\sum \left({\overset{\ldots }{S_{x}}}^{2}+{\overset{\ldots }{S_{z}}}^{2}\right)$$
(7)$$J_{A}=\sum {\overset{\ldots }{S_{y}}}^{2}$$

Here,

$J_{l},\;J_{r}$: Linear and angular componets of jerk cost function

$\overset{\ldots }{S_{x}},\;\overset{\ldots }{S_{z}}$: Linear jerk along the x-and z-axes

$\overset{\ldots }{S_{y}}$: Angular jerk around the y-axis

Several studies on robot manipulation have reported that a smooth low-jerk motion can reduce actuator load and wear [45]. Additionally, a smoother motion with a smaller jerk can be tracked faster and more accurately [46]. As one of many theories on which human motor control is based, the minimum-jerk hypothesis suggests that humans minimize jerk in skillful limb movements [47]. In this study, balancing movements with lower jerk were interpreted as being more skillful movements.

#### 2.4.2. Experimental Tests with Actual Robot

In the experimental tests using the actual robot, the motion of the robot trunk was confined to the frontal plane using kinematic constraints. External forces were manually applied to random locations on the trunk of the robot. Owing to the nature of manual force applications, the force could not be exerted in a systematic manner with controlled amplitudes, patterns, and durations. To address this problem, the integral of the force over time or the impulse was monitored instead of the amplitude of the force, while the magnitude of the force was controlled considering the simulation results. The resistance to balance loss was assessed by monitoring the impulse at which either foot slip or loss of foot-ground contact occurred. The loss of foot-ground contact was detected when the vertical displacement of one foot of the bipedal robot exceeded a threshold. In the experiments using the actual robot, a higher impulse at which balance loss occurred was interpreted as greater resistance to balance loss. The jerk of the balancing motion was monitored and interpreted in the same way as in the experiment using the dynamic simulator.

## 3. Results and Discussion

### 3.1. Selected DNN Structures Based on Performance Comparison

With the hyperparameters described in the previous section, the three DNN structures for *DCM Net* and *W Net* were optimized. The performances of the optimized DNN structures are compared with respect to RSME and computation time in Table 2 and Table 3. The RMSEs for the DCM and the net wrench indicate the goodness of model fit to the human-operated balancing data in the frontal plane (two linear components (x- and z-axes) and one angular component (y-axis)). Based on the performance analysis, the DNN structures for *DCM Net* and *W Net* were selected.

#### 3.1.1. Comparison of DNN Structures for Robot Dynamic Simulator

By using a test data set that was not used for training, the performance of the three candidates for the DNN structures was evaluated with the robot dynamic simulator. Figure 9 compares the predictions from the three optimized DNN structure with the target output. Figure 9a–c plots the outputs from *DCM Net*, and Figure 9d–f plots the outputs from *W Net*. In the figures, the grey, green, blue, and red lines represent the target output and the predictions from MLP, CNN-MLP, and LSTM-MLP, respectively. As can be seen in the figure, the predictions from the three DNN structures are very close to the target output, which makes it difficult to discern the performance of the three candidates for the DNN structures.

The RMSE and computation time of the three DNN structures after optimization are compared in Table 2 to determine the DNN structure for *DCM Net* and *W Net*. The MLP structure was selected for the robot dynamic simulator as the DNN structure for *DCM Net*. The results show that the RMSEs had comparable accuracies for all three optimized DNN structures for *DCM Net*. In terms of the computation time, the LSTM-MLP structure showed the slowest computation time, whereas the MLP structure showed the fastest computation time for *DCM Net*. The MLP structure was selected as the DNN structure for *W Net* for the robot dynamic simulator. Although all three candidates for *W Net* had comparable accuracy, the MLP structure had a much shorter computation time compared with the other two.

The optimized MLP structures for *DCM Net* and *W Net* are described in Figure 10. Both *DCM Net* and *W Net* employ MLP structures in the hidden layer with different sets of hyperparameters. A 2D array of 9 by 100 was fed into both *DCM Net* and *W Net*. Since the MLP structure uses a 1D vector as an input, *DCM Net* and *W Net* convert the 2D input array into a 1D vector of size 900 using a flatten layer. *DCM Net* and *W Net* have six and five fully connected (FC) layers, respectively. *DCM Net* uses ReLu activation function for its six FC layers and a linear activation function for its output layer, whereas *W Net* uses Tanh activation functions for all FC layers and the output layer. During the training process, the Adam optimizer was used to update the weights of *DCM Net* and *W Net*. The learning rate (LR), the decay rate of the LR (DR), and the batch size (BS) were set to 0.000719, 0.005219, and 1024 for *DCM Net* and 0.002875, 0.009715, and 128 for *W Net*, respectively.

#### 3.1.2. Comparison of DNN Structures for the Actual Robot

The three candidates for the DNN structures were evaluated with the actual robot using a test dataset that was not used for training. Figure 11 shows the target output and the predictions from the three optimized DNN structure. The grey, green, blue, and red lines in the figure denotes the target output and the predictions from MLP, CNN-MLP, and LSTM-MLP, respectively. Figure 11a–c compares the outputs from *DCM Net*, and Figure 11d–f compares the outputs from *W Net*. 

The evaluation results are listed in Table 3. The CNN-MLP structure was selected as the DNN structure for *DCM Net* to estimate the reference DCM ($\xi_{ref}$) from the CoM states of the actual robot. The CNN-MLP structure showed better accuracy than the other two candidates. The MLP structure showed the fastest computation time, although its accuracy was much lower than that of CNN-MLP. The computation time of the CNN-MLP structure was sufficiently fast because it was still faster than the control-loop execution time of the actual robot. The optimized CNN-MLP structure was selected as the DNN structure for *W Net* to estimate the reference wrench ($W_{ref}$) from the CoM states of the actual robot. Although all the candidates for DNN structures showed comparable accuracy, the CNN-MLP structure demonstrated slightly better overall accuracy. The CNN-MLP structure was also superior to the other two candidates with respect to the computation time. For both *DCM Net* and *W Net*, the computation time of the LSTM-MLP structure was much slower than the control-loop execution time of the actual robot, which makes it inappropriate for actual robot applications.

The CNN-MLP structures for *DCM Net* and *W Net* are illustrated in Figure 12. Both *DCM Net* and *W Net* used CNN-MLP structures in the hidden layer with different sets of hyperparameters. A 2D array of robot states was fed to both *DCM Net* and *W Net*. *DCM Net* has two pairs of convolution-pooling (C-P) layers, whereas *W Net* has four pairs of C-P layers. In both networks, the output of the last C-P layer was transformed into a 1D vector using a flatten layer to be later used for the FC layer. Both *DCM Net* and *W Net* have nine FC layers after the last C-P layer. During the training process, the RMSprop optimizer was used to update the weights of *DCM Net* and *W Net*. The LR, DR, and BS were set to 0.0001, 0.000001, and 1024 for *DCM Net* and 0.0001, 0.000001, and 1024 for *W Net*, respectively.

### 3.2. Performance of Balance Controller in Robotic Dynamic Simulator

In the robot dynamic simulator, the performance of the balance controller using the behavior cloning model (*BC_BC_*) was compared with that of the conventional control based on the FRC (*BC_FRC_*). With the robot dynamic simulator, the pattern of the external force, including the amplitude, pulse width, and time interval, can be easily modified. The performances of *BC_BC_* and *BC_FRC_* were evaluated in terms of the resistance to balance loss and the jerkiness of the balancing motion.

#### 3.2.1. Resistance to Balance Loss

The resistance to balance loss was evaluated by the amplitude of the external force at which either foot slip or loss of foot-ground contact occurred. Figure 13 compares the force amplitudes between *BC_BC_* and *BC_FRC_* at which the foot slip (blue bars) and the loss of foot-ground contact (red bars) occurred when a single pulse (Figure 13a) and a series of pulses (Figure 13b) of external forces with widths of 180, 150, 100, 50, and 10 ms were applied. As can be seen in the figure, both foot slip and loss of foot-ground contact occurred at higher force amplitudes with *BC_BC_* than with *BC_FRC_*, which demonstrates superior resistance to balance loss with the behavior cloning model. The simulation results show that the robot under the control of the behavior cloning model performed better than the conventional FRC-based control in maintaining balance against a series of square pulses, as well as a single square pulse with various pulse widths.

#### 3.2.2. Jerkiness of Balancing Motion

The jerkiness of the balancing motion was evaluated based on the jerk cost function. The jerk cost functions of the linear and angular movements were computed using Equations (6) and (7) when a single pulse was applied with varying amplitudes from 21 to 27 N and a pulse width of 100 ms. Table 4 compares the jerk cost functions with *BC_BC_* and *BC_FRC_* during the application of a single square pulse.

The results show that the jerk cost functions with *BC_BC_* were lower than those with *BC_FRC_* for both linear and angular balancing movements in the horizontal plane for the pulse amplitudes of 21 N and 22 N. Particularly for the angular motion, the jerk cost functions with *BC_BC_* were lower than those with *BC_FRC_* by an order of magnitude. As described in the third column of Figure 13a, the robot controlled by *BC_FRC_* lost its balance with a pulse amplitude above 23 N, whereas with *BC_BC_*, the balance loss occurred with a pulse amplitude of 28 N. The simulation results show that the robot controlled by the behavior cloning model generated smoother balancing movements for higher external forces without losing its balance, compared to the conventional FRC-based control. This feature is advantageous for improving the lifespan of the motors and other mechanical parts of the robot by reducing jerky impact loads.

### 3.3. Performance of Balance Controller in Actual Robot

The performances of *BC_BC_* and *BC_FRC_* were tested with an actual robotic system, Little HERMES. For experimental tests with the actual robot, external forces with irregular shapes were manually applied as the disturbance to balance. The performances of *BC_BC_* and *BC_FRC_* were evaluated in terms of the resistance to balance loss and the jerkiness of the balancing motion.

#### 3.3.1. Resistance to Balance Loss

In the robot dynamic simulator, the resistance to balance loss was evaluated using the amplitude of the external force at the balance loss. In experimental tests with the actual robot, we used the impulse (the time integral of the external force) with which foot-ground contact was lost, rather than the external force itself, to compare the performance of *BC_BC_* and *BC_FRC_*. The impulse is a more suitable measure than the force since the magnitude of the manually applied force is difficult to control.

In the experiments, impulsive forces were manually applied to the trunk of the robot. The impulse of the force was increased while its magnitude was monitored until foot-ground contact was lost. We performed 10 sets of experiments, and each set of experiments included five cases of foot-ground contact loss. The minimum value of impulse out of the five cases was selected as the threshold impulse that initiated foot-ground contact loss for each set of experiments. From the 10 sets of experiments, 10 threshold impulses were selected.

The bar graph in Figure 14 compares the mean and standard deviation values of the threshold impulses with *BC_BC_* and *BC_FRC_*. The results show that the threshold impulses with *BC_BC_* (mean = 2649.3, std = 178.4) was much higher than those with *BC_FRC_* (mean = 1982.6, std = 177.9), with a statistically significant difference (*p* < 0.05). These results further support the fact that the behavior-cloning model can improve the capability of the robot to maintain balance compared to the *BC_FRC_* against irregularly shaped external forces.

#### 3.3.2. Jerkiness of Balancing Motion

The jerk cost functions for the linear and angular movements were computed for the balancing motion of the actual robot. Figure 15 compares the averages of the jerk cost functions for the linear and angular movements controlled by *BC_BC_* (blue bars) and *BC_FRC_* (red bars). In the figure, the averages of the jerk functions were compared in four different ranges of the impulse magnitude: [1500, 2000), [2000, 2500), [2500, 3000), and [3000, 3500). The robots equipped with *BC_BC_* and *BC_FRC_* were able to maintain its balance with impulse magnitudes up to 3500 N·s and 2500 N·s, respectively. For the jerk cost function of linear motion, *BC_BC_* showed slightly larger cost values than *BC_FRC_* in the ranges of [1500, 2000) and [2000, 2500). As for the jerk cost function of angular motion, however, *BC_BC_* showed lower cost values than did *BC_FRC_* for the same ranges. The experimental results show that the behavior cloning model can generate angular motion with higher smoothness, and linear motion with comparable smoothness, compared to the conventional FRC-based control.

## 4. Conclusions

In this study, we developed a novel balance controller for bipedal robots in the frontal plane by employing a behavior cloning model, as one of deep learning techniques. Unlike FRC-based studies on robot balancing, the developed control algorithm generates the desired wrench at the CoM by capturing human balancing behavior from the robot states without referring to a specific reference frame.

The developed balance controller consists of two components: a wrench estimator and a joint torque controller. The wrench estimator calculates the desired wrench at the CoM using a pre-trained behavior cloning model that employs two separate DNNs. The two DNNs, *DCM Net* and *W Net*, were trained on human-operated balancing data acquired using the BFI. The two DNN structures after training the map from the measured robot states to the desired wrench at the CoM of the robot. The joint torque controller calculates the joint torques from the desired wrench based on the robot dynamics model of the bipedal robot. This structure allows the developed balance controller to cope with various contact conditions, including changing the contact points and time-varying contact forces.

The performance of the developed system was evaluated experimentally using an actual robotic system, as well as in simulations using a robot dynamics model. Both the simulation and experimental results show that the developed controller outperforms conventional FRC-based balance controllers with respect to the prevention of foot slip and foot contact loss under various types of external perturbations. The results also show that the developed balance controller can generate smoother angular movements compared to conventional FRC-based controllers. These results confirm that the balancing controller developed in this study outperforms the conventional FRC-based balance controller by moving the reference configuration based on human balancing movements, while employing the same PD-controller as in the conventional balancing method.

As a preliminary study, we investigated the balancing control of a bipedal robot in the frontal plane. Although balancing in the sagittal plane involves different balancing mechanisms, the behavior cloning-based control methodology developed for frontal plane motion can be easily extended to achieve control balance in the sagittal plane. In our future work, we plan to apply the balance controller to a bipedal robot performing power manipulations in the sagittal plane, such as hammering and axing.

## Figures and Tables

**Figure 1 biomimetics-07-00232-f001:**
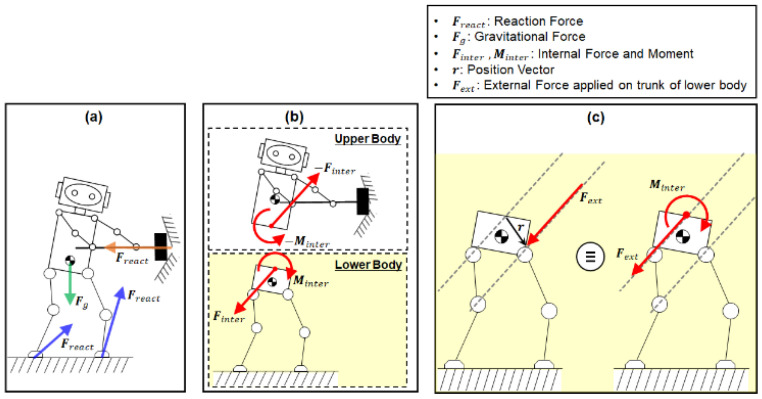
Two parallel-coupled sub-systems of the humanoid: (**a**) Free body diagram of the humanoid interacting with the environment, (**b**) Internal force and moment between the upper and lower bodies, (**c**) External force having the same effect as the internal force and moment.

**Figure 2 biomimetics-07-00232-f002:**
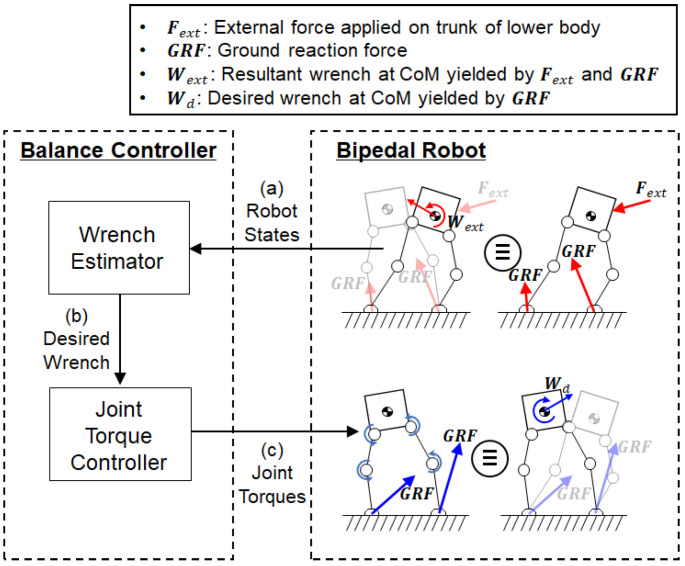
Overview of the balance control system: A wrench estimator uses (**a**) the CoM states of the bipedal robot to predict (**b**) the desired wrench at the CoM. A joint torque controller calculates (**c**) the joint torques required to achieve the desired wrench at the CoM.

**Figure 3 biomimetics-07-00232-f003:**
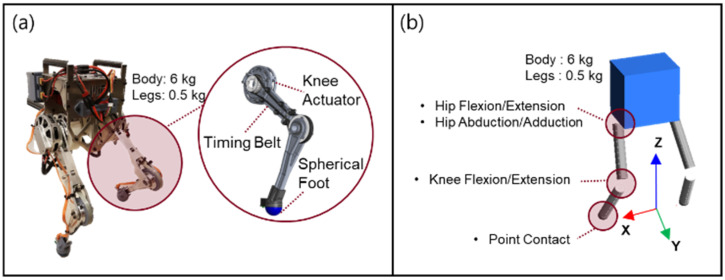
Bipedal robot platform used to acquire the human-operated balancing data and evaluate the performance of the developed balance control system: (**a**) Actual robot, Little HERMES. (**b**) Multi-body dynamics model of Little HERMES programmed via physics engine, MuJoCo.

**Figure 4 biomimetics-07-00232-f004:**
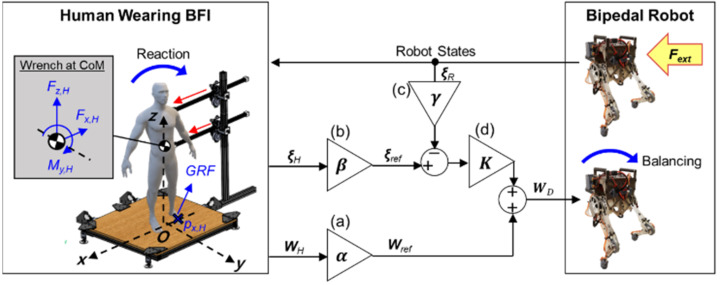
Human-operated balancing system using BFI: Desired wrench at the CoM of the robot (*W**_D_*) was computed using net wrench applied at the CoM of the human operator (*W**_H_*), divergent components of motion of the robot and the human operator (*ξ**_R_* and *ξ**_H_*) while feedback force was exerted on human’s torso referring to measured robot states. Here, scale factors (**a**) $\alpha ={\left[\alpha_{x}\;\alpha_{z}\;\alpha_{y}\right]}^{T}$ for *W**_H_* (**b**) $\beta ={\left[\beta_{x}\;\beta_{z}\;\beta_{y}\right]}^{T}$ for *ξ**_H_*, and (**c**) $\gamma ={\left[\gamma_{x}\;\gamma_{z}\;\gamma_{y}\right]}^{T}$ for *ξ**_R_* and control gains (**d**) $K={\left[K_{x}\;K_{z}\;K_{y}\right]}^{T}$ in Equation (3) were used to convert the human reaction to *W**_D_*.

**Figure 5 biomimetics-07-00232-f005:**
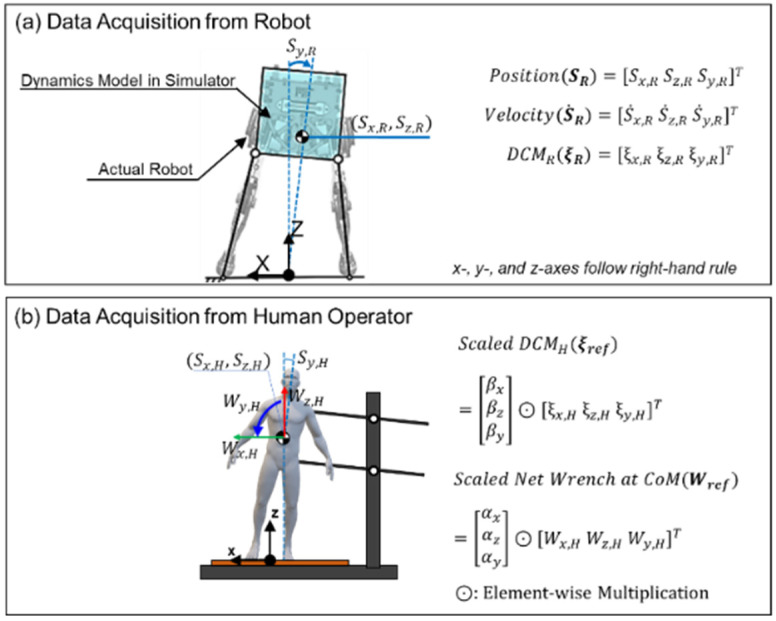
Collected data used to train the balance controller: (**a**) Position, velocity, and DCM collected from the bipedal robot, (**b**) DCM and net wrench data collected from the human operator, which were scaled as shown in Equation (3) to be used for training data.

**Figure 6 biomimetics-07-00232-f006:**
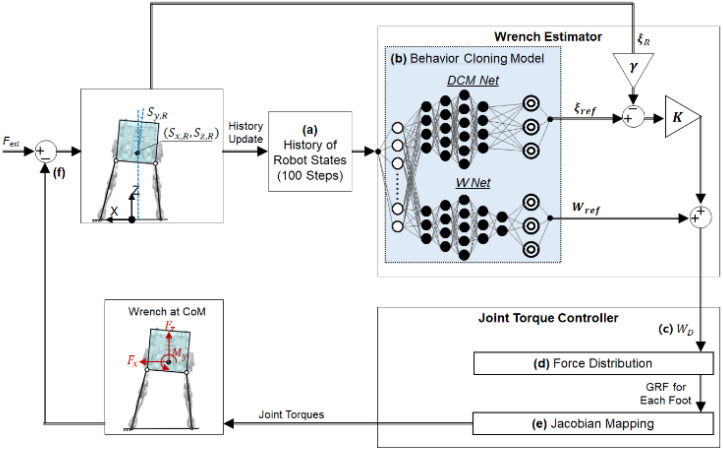
Schematic diagram for the control process of the robot balancing: (**a**) At each control loop, the oldest robot state is removed, and the current robot state is inserted to update the data to be provided to the wrench estimator. (**b**) The behavior cloning model with two separate networks: *DCM Net* and *W Net*. The two networks use the same input data of robot states but yield different outputs: $\xi_{ref}$ and $W_{ref}$, respectively. (**c**) The desired wrench at the CoM ($W_{D}$ ) is computed from the outputs of the behavior cloning model. (**d**) GRF of each foot is computed from desired wrench at CoM. (**e**) Joint torques are computed from the GRF using the Jacobian matrix of each leg. (**f**) GRFs from both feet generate the net wrench at the CoM to maintain balance.

**Figure 7 biomimetics-07-00232-f007:**
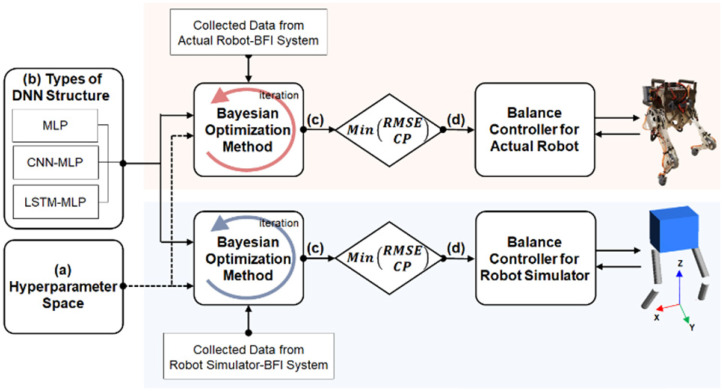
Optimization process for the hyperparameters of *DCM Net* and *W Net*. The Bayesian optimization method iteratively explores (**a**) types and range of hyperparameters for (**b**) each of the three types of DNN structures. (**c**) The three types of DNN structures were optimized to lower the RMSE between the predictions from the DNN and the target values collected via the BFI system. (**d**) The performance of the optimized MLP, CNN-MLP, and LSTM-MLP were compared to select the best structure for the balance controller by considering the RMSE and the computation time (CP).

**Figure 8 biomimetics-07-00232-f008:**
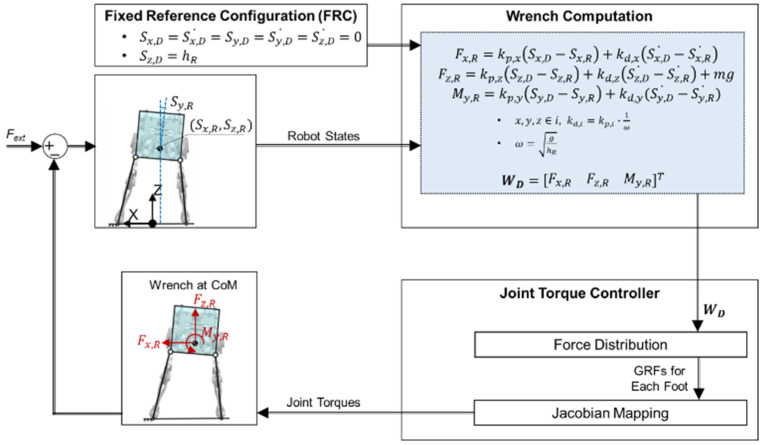
Conventional balance controller using the FRC: Desired wrench at CoM is calculated by simple feedback laws referring to the FRC, unlike the developed balance controller which uses a behavior cloning model.

**Figure 9 biomimetics-07-00232-f009:**
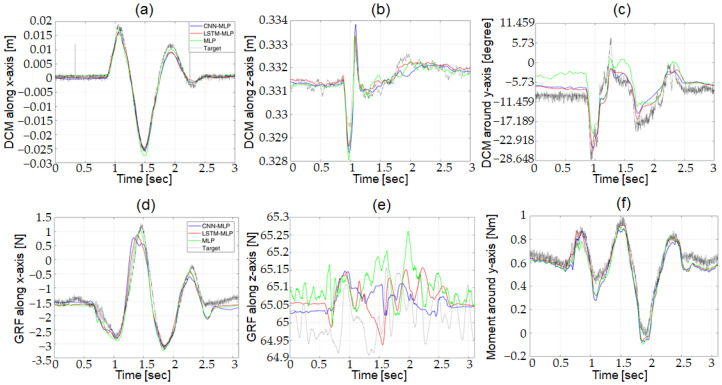
Comparison of target output and predictions from three optimized DNN structures using test data in robot dynamic simulator; (**a**) DCM along x-axis, (**b**) DCM along z-axis, (**c**) DCM around y-axis, (**d**) GRF along x-axis, (**e**) GRF along z-axis, and (**f**) Moment around y-axis.

**Figure 10 biomimetics-07-00232-f010:**
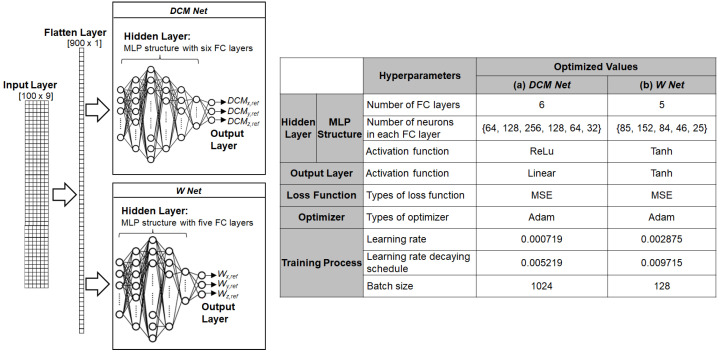
Structures of *DCM Net* and *W Net* for the behavior cloning model used in the dynamic simulator: hyperparameters for (**a**) *DCM Net* and (**b**) *W Net.*

**Figure 11 biomimetics-07-00232-f011:**
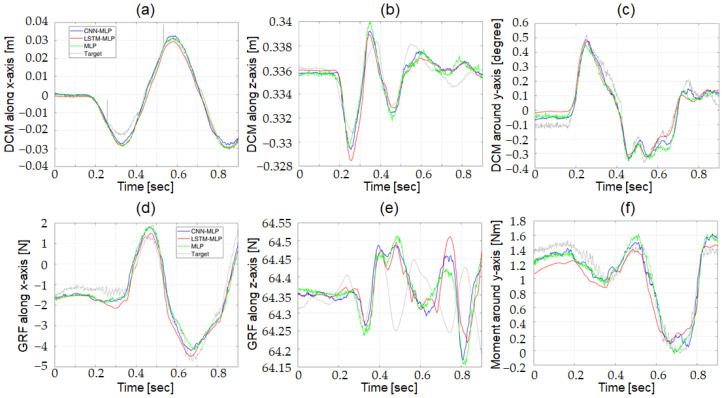
Comparison of target output and predictions from three optimized DNN structures using test data in actual robot; (**a**) DCM along x-axis, (**b**) DCM along z-axis, (**c**) DCM around y-axis, (**d**) GRF along x-axis, (**e**) GRF along z-axis, and (**f**) Moment around y-axis.

**Figure 12 biomimetics-07-00232-f012:**
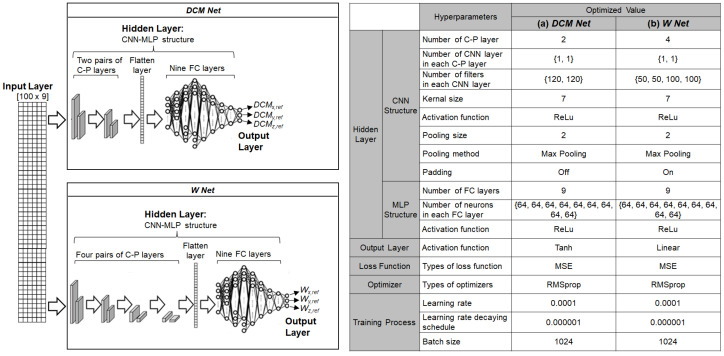
Structures of *DCM Net* and *W Net* for the behavior cloning model used in actual robot: hyperparameters for (**a**) *DCM Net* and (**b**) *W Net.*

**Figure 13 biomimetics-07-00232-f013:**
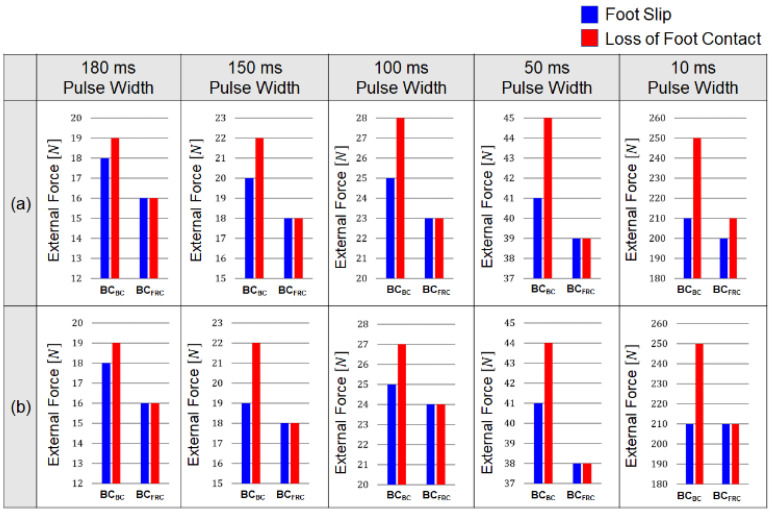
Amplitude of the external forces where foot position of the robot in the dynamic simulator changed in the lateral direction (foot slip) and the vertical direction (loss of foot contact): Results when a square-shaped external force with a pulse width of 180, 150, 100, 50, and 10 ms was applied with the inter pulse interval of (**a**) 3600 ms and (**b**) 400 ms.

**Figure 14 biomimetics-07-00232-f014:**
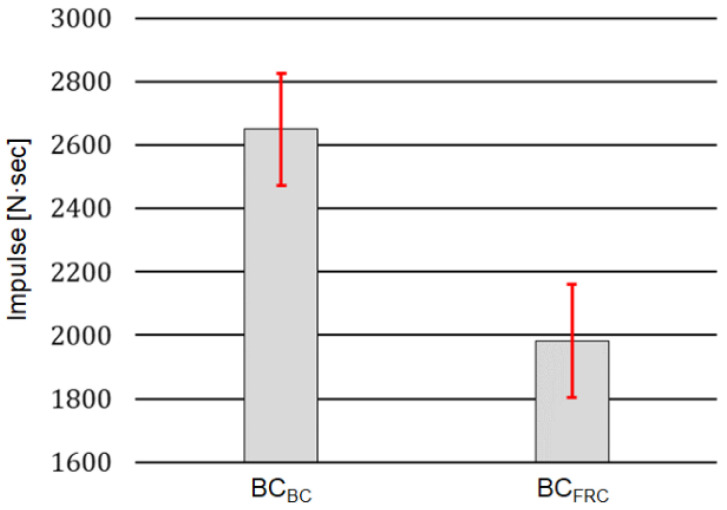
Mean and standard deviation values for the 10 threshold impulses observed from the robot controlled by *BC_BC_* and *BC_FRC._*

**Figure 15 biomimetics-07-00232-f015:**
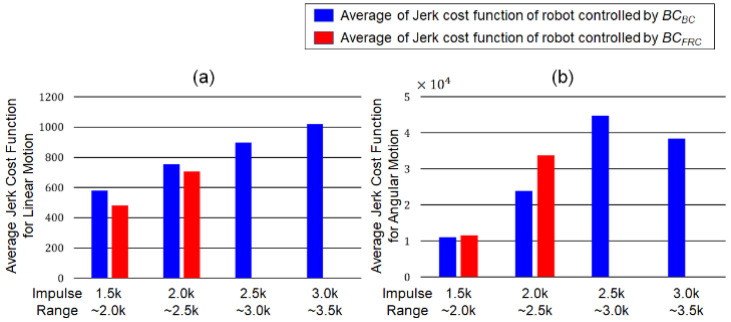
Averages of the jerk cost functions calculated from the balancing motion under the external force whose impulse belongs to each sub-range: [1500, 2000), [2000, 2500), [2500, 3000), and [3000, 3500). Blue and red bars representing the averages of the jerk cost functions calculated from the robots controlled by *BC_BC_* and *BC_FRC_*, respectively. (**a**) Average of the jerk cost functions for linear movement, (**b**) Average of the jerk cost functions for angular motion.

**Table 1 biomimetics-07-00232-t001:** Explored hyperparameters and their ranges.

		Hyperparameters	Explored Range
Hidden Layer	MLP Structure	Number of fully connected layers	Integers
		Number of neurons in each fully connected layer	Integers
		Activation function for neurons	{Tanh, ReLu, Linear}
	CNN Structure	Number of CNN-pooling Layers	Integers
		Number of CNN layers in each CNN-pooling layer	{1, 2, 3}
		Number of filters in each CNN layer	Integers
		Kernel size	{3, 5, 7}
		Activation function for CNN layers	{Tanh, ReLu, Linear}
		Pooling size	{2, 3, 4, 5}
		Pooling method	{Max, Average}
		Padding	{On, Off}
	LSTM Structure	Number of LSTM cells in each LSTM layer	Integers
		Number of LSTM layers to be stacked	{1, 2, 3}
Output Layer		Activation function	{Tanh, ReLu, Linear}
Loss Function		Types of loss function	{MSE, Hubber, Log-Cosh}
Optimizer		Types of optimizers	{SGD, Adam, RMSprop}
Training Process		Learning rate	Real Numbers
		Learning rate decaying schedule	Real Numbers
		Batch size	Integers

**Table 2 biomimetics-07-00232-t002:** Performance of Three Optimized DNN Structures for *DCM Net* and *W Net* of the Robot Simulator.

Type of DNN	Optimized DNN Structure	RMSE in x-Axis	RMSE in z-Axis	RMSE in y-Axis	Computation Time
*DCM Net*	MLP	0.0081 m	0.0042 m	0.0026°	0.235 ms
	CNN-MLP	0.0091 m	0.0042 m	0.0021°	0.316 ms
	LSTM-MLP	0.0072 m	0.0041 m	0.0021°	1.171 ms
*W Net*	MLP	0.00057 N	0.0019 N	0.0033 Nm	0.11 ms
	CNN-MLP	0.0054 N	0.0019 N	0.0032 Nm	0.713 ms
	LSTM-MLP	0.0055 N	0.0019 N	0.0031 Nm	4.153 ms

**Table 3 biomimetics-07-00232-t003:** Performance of Three Optimized DNN Structures for *DCM Net* and *W Net* of the actual robot.

Type of DNN	Optimized DNN Structure	RMSE in x-Axis	RMSE in z-Axis	RMSE in y-Axis	Computation Time
*DCM Net*	MLP	0.0113 m	0.0046 m	0.0033°	0.0755 ms
	CNN-MLP	0.011 m	0.0044 m	0.003°	0.5104 ms
	LSTM-MLP	0.0132 m	0.0044 m	0.0028°	2.5324 ms
*W Net*	MLP	0.0095 N	0.0018 N	0.0144 Nm	0.767 ms
	CNN-MLP	0.0093 N	0.0018 N	0.0124 Nm	0.2918 ms
	LSTM-MLP	0.0096 N	0.0018 N	0.0112 Nm	14.2904 ms

**Table 4 biomimetics-07-00232-t004:** Jerk cost functions with different amplitudes of pulses.

Jerk Cost Function		Amplitude of External Force						
		21 N	22 N	23 N	24 N	25 N	26 N	27 N
Linear Component	*BC_BC_*	61.4	75.5	88.2	99.1	106.7	103.6	113.1
	*BC_FRC_*	75.1	134.5	Loss of Foot Contact				
Angular Component	*BC_BC_*	762.8	2595.4	4303.8	5356.3	5902.7	5749.5	7476.3
	*BC_FRC_*	4073.9	20,555	Loss of Foot Contact				

## Data Availability

Not applicable.

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
