# Peer review of "Standing Balance Control of a Bipedal Robot Based on Behavior Cloning"

_biomimetics, 2022, doi:10.3390/biomimetics7040232_

Round 1

Reviewer 1 Report

This paper proposes a standing balance controller for humanoids based on two artificial neural networks trained with data extracted from the balancing mechanism of actual humans. The proposed controller is tested and compared with a commonly used balancing strategy in a simulated and a real robot with good results. The paper is easy to read, it presents a proper structure, and it is clearly explained. Nevertheless, this reviewer has detected some minor issues:

1. In eq. (4) the authors show how the scale factors are obtained, but they only indicate the value of the control gain. How can the control gain be computed?

2. The description of the variables involved in equations (4), (5) and (7) should not be in italics

3. The number of equation (3) is in italics and (5) in bold

4. There is a wrong subscript in lines 624-625 "BC_{BC} showed lower average values than did BC_{BC} for the same ranges."

5. Although the paper is clearly written, there are some sentences that could be improved, e.g., in line 250 it is stated "Equation (4) summarizes the computation of the desired wrench for a bipedal robot (W_D) to maintain the balance of the bipedal robot".

6. Figure 5 is on a different page than its caption.

Author Response

Please see the atachment. Thank you.

Reviewer 2 Report

The paper is well written.

It will be good if the authors can provide a video of the simulation and real world experiment to enhance the quality of this paper/work. 

Human is involved in this study. Authors will need to clearly indicate the ethical clearance approval for this study. The work has to comply with the standard convention. 

There are variations in human movement, particularly in how human body responds to the external perturbation. How does the proposed system handle this variation? 

Will the basic anthropometric properties e.g. length of hand, weight, age, height, gender, height of the shoulder, position of COM, etc of the human operator affect the performance of the robot? Please elaborate on this. 

It will be better if authors can provide comparison with the existing balancing method used by the bipedal robot. Because as of now, it is quite hard to comprehend why behavior cloning is required when the conventional balancing method used by the bipedal robot can achieve good results and is behavior cloning is more superior than the conventional one? 

Authors are suggested to provide time based graph(s) to illustrate the dynamic behavior of the robot and human operator during the balancing routine, instead of just the rmse, and amplitude. 

Author Response

Please see the atachment. Thank you.

Round 2

Reviewer 2 Report

NA
